# TIER Version 1.0: An open-source Topographically InformEd Regression (TIER) model to estimate spatial meteorological fields

Andrew J. Newman[1] and Martyn P. Clark[2]

[1]Research Applications Laboratory, National Center for Atmospheric Research*, Boulder CO, 80307-3000, USA

[2]University of Saskatchewan, Centre for Hydrology and the Global Institute for Water Security, Saskatoon, SK

*The National Center for Atmospheric Research is sponsored by the National Science Foundation

*Correspondence to*: Andrew Newman (anewman@ucar.edu)

**Abstract.** This paper introduces the Topographically InformEd Regression (TIER) model, which uses terrain attributes in a regression framework to distribute *in situ* observations of precipitation and temperature to a grid.  The framework enables

our understanding of complex atmospheric processes (e.g. orographic precipitation) to be encoded into a statistical model in an easy to understand manner.  TIER is developed in a modular fashion with key model parameters exposed to the user. This enables the user community to easily explore the impacts of our methodological choices made to distribute sparse, irregularly spaced observations to a grid in a systematic fashion. The modular design allows incorporating new capabilities in TIER.  Intermediate processing variables are also output to provide a more complete understanding of the algorithm and

any algorithmic changes.  The framework also provides uncertainty estimates. This paper presents a brief model evaluation and demonstrates that the TIER algorithm is functioning as expected.  Several variations in model parameters and changes in the distributed variables are described.  A key conclusion is that seemingly small changes in a model parameter result in large changes to the final distributed fields and their associated uncertainty estimates.

## 1 Introduction

Gridded near-surface meteorological products (specifically precipitation and temperature) are a foundational product for many applications including weather and climate model validation, hydrologic modeling, climate model downscaling, among others (Day 1985; Franklin 1995; USBR 2012; Pierce et al. 2014; Liu et al. 2016).  It is often challenging to develop realistic estimates of these variables, particularly when complex terrain or large spatial climate gradients are present in the domain of interest.  Because of their widespread usage and potential challenges generating products, a plethora of methods

have been developed ranging from nearest neighbors, distance weighted interpolation, Kriging, knowledge-based, climatologically aided interpolation, Gaussian filters, multiple linear regression, and others (Thiessen 1911; Shepard 1968,

1984; Chua and Bras 1982; Daly et al. 1994; Willmott and Roebson 1995; Thornton et al. 1999; Banerjee et al. 2003; Clark and Slater 2006; Cressie and Wikle 2011; Nychka et al. 2015, Cornes et al. 2018).

Across the methods, nearest neighbor and distance weighted interpolations use spatial distance as the only predictor, a reasonable choice in areas with high station densities, but much less so in sparsely gauged regions. The resultant field is also discontinuous between station areas of influence for nearest neighbor, while distance weighted interpolation will increase precipitation occurrence unless explicit occurrence prediction is included (Thornton et al. 1997; Newman et al. 2015, 2019). Climatologically aided interpolation assumes the climatological field is better resolved by the available observations and has a strong relationship with the field of interest (e.g. daily precipitation) such that using the climatological field in the final interpolation increases the output information content (Willmot and Robeson 1995). These assumptions are invalid when the climatological field is poorly resolved, or has little correspondence to the field of interest which happens when an event has a significantly different pattern than climatology (e.g. Lundquist et al. 2015; Newman et al. 2019). Kriging and linear regression frameworks may include multiple spatial predictors and uncertainty estimates. However, these methods may also produce unrealistic results with sparse station observations (Cornes et al. 2018). Finally, knowledge based systems impose a regularization on the input data through knowledge-based rules (section 2). This allows for physically plausible interpolation fields in sparsely gauged regions, but inflexibility similar to climatologically aided interpolation. Finally, in sparsely observed regions we do not know the true error characteristics of any method.

The currently available climate products that use these methods have complex processing systems (Daly et al. 2008; Xia et al. 2012; Livneh et al. 2015; Newman et al. 2015; Thornton et al. 2018). The product workflow typically includes many processing steps, methodological choices, and model parameters, all interacting to influence the characteristics of the final product. Therefore, comparison studies of product performance (and even single product evaluations) are often not able to attribute differences at the final output level to specific methodological choices (Newman et al. 2019). To help alleviate these difficulties and improve our understanding of method performance across conditions, flexible modular software systems need to be developed that expose model parameters to the users and allow for new functionality to be easily added (Clark et al. 2011).

This paper focuses on approaches that incorporate knowledge of atmospheric physics into relatively simple underlying statistical models (e.g. orographic precipitation, temperature lapse rates into linear regression models) to improve the accuracy of the gridded field (e.g. Daly et al. 1994; Willmott and Matsuura 1995). Daly et al. (1994), hereafter D94, develop a complex knowledge-based system consisting of: 1) terrain preprocessing; 2) station selection; 3) development of a locally weighted meteorological variable-elevation linear regression; and 4) postprocessing. Omitted here are the pre-processing steps to screen station data (including quality control), and filling missing or suspicious data values. Following D94, many studies have included new knowledge-based capabilities, new intermediate processing steps, and increased granularity in a

given step (Daly et al. 2002, 2007, 2008). In recent papers there are upwards of 15-20 model parameters that have varying degrees of influence on the final product.

The Topographically InformEd Regression (TIER) model implements the knowledge-based approach described in D94 and subsequent papers (Daly et al. 2000, 2002, 2007, 2008, hereafter D00, D02, D07, and D08). Note that TIER is not designed to be an exact replica as it does not match any source code, nor does it implement all features described in D94, D00, D02, D07, and D08. The paper is organized as follows: we introduce the TIER conceptual model in section 2.1, the preprocessing algorithms in section 2.2, the regression model in section 2.3, and post-processing routines in section 2.4. Then, a brief model evaluation is included in section 3 to verify that the TIER model is functioning as expected. Next, we explore model parameter variation experiments for three simple test cases to highlight how model parameter choices impact the final product in section 3.1. Finally, a summary discussion of TIER, lessons learned from the parameter experiments, and next steps are discussed in section 4 with code and data availability in sections 5 and 6 respectively.

## 2 TIER Methodology

### 2.1 Conceptual model

Precipitation and temperature are unevenly distributed around the globe for myriad reasons including general circulation patterns and landscape effects. Following D94, TIER assumes that large-scale gradients are resolved by the input station data, and incorporates direct knowledge of atmospheric physics to account for landscape effects (e.g., orographic precipitation, mesoscale circulations near the coast). Since the landscape influences the distribution of precipitation and temperature, particularly their climatology, many past studies have developed methods to use terrain attributes to estimate meteorological fields (e.g. Spreen 1947; Phillips et al. 1992). For example, orography has a particularly strong influence on precipitation by enhancing uplift of air (e.g. Schermerhorn 1967; Smith and Barstad 2004).

D94 develops a method to use a high-resolution DEM to produce empirical estimates of the precipitation-elevation relationship. They demonstrate that using actual station elevations in the precipitation-elevation relationship leads to a weak or nonexistent relationship, while using a coarse resolution DEM smooths out local variability and results in a stronger relationship between precipitation and elevation. Such stronger relationships occur because microscale terrain features have a much smaller impact on the atmosphere than the larger-scale terrain features of the order of 2-15km (D94 and references therein). Of course, the optimal length scale varies across atmospheric conditions and for each precipitation event, but in general a coarse resolution or smoothed high-resolution DEM provides a strong basis for developing robust climatological precipitation-elevation relationships. Additionally, the amount of precipitation varies according to aspect (e.g. windward or lee slope), suggesting the need for different relationships for different aspects (e.g. Alter 1919; Houghton 1979). D94 use

the smoothed DEM, and decompose the domain into directional "facets" that all individually have a separate precipitation-elevation relationship. Facets are defined as continuous areas with similar aspects (slope orientation).

Daly and his colleagues have introduced several methodological enhancements since the seminal D94 paper. D00 expand on
D94 to include maximum and minimum temperature while D02 fully describes the knowledge-based system and the various
physical processes included in it. Beyond elevation, the influence of large bodies of water on precipitation and temperature
are incorporated by using coastal proximity, or distance to the coastline. Finally, cold air drainage down slopes and
subsequent pooling in valleys is modeled as well. A conceptual two-layer atmosphere where layer 1 is the boundary layer
containing temperature inversions and layer 2 is the free atmosphere is applied to the DEM. A simple two-layer atmospheric
model for temperature is necessary to capture near surface temperature inversions. This method identifies areas highly
susceptible to inversions (e.g. valleys) and allows for the model to have temperature lapse rate reversals from increasing
temperature with height below to decreasing above the inversion. Grid points that are identified to be within the boundary
layer (layer 1) are allowed to have strong temperature inversions. These grid points are identified using the topographic
position concept, which essentially computes the D94, D02, and D08 provide extensive details on the underlying theory of
this knowledge-based approach.

### 2.2 TIER Terrain preprocessing

The TIER preprocessing routines consist of the functions used to generate the required terrain attributes for the regression
model. Currently, this consists of functions that perform netCDF input/output (IO), process the DEM into topographic
facets, the distance to the coast, topographic position, and estimate the idealized two-layer atmosphere. A parameter and
control file specifies model parameters, and IO directories and files; see Tables 1 and 2 respectively. A flow chart describing
the general flow, order of operations, and data requirements is given in Figure 1.

### 2.2.1 Topographic facets

The native resolution DEM is first smoothed using a user-defined filter (Table 2). The 'Daly' filter is defined as (D94):

$$S_{E\,i,j} = \frac{1}{2}E_{i,j} + \frac{1}{8}\left(E_{i+1,j} + E_{i-1,j} + E_{i,j+1} + E_{i,j-1}\right) \tag{1}$$

where $S_{E\,i,j}$ is the smoothed elevation and $E_{i,j}$ is the high-resolution elevation at grid point (i, j). D94 computes multiple
smoothed DEMs to account for data density changes across the domain while TIERv1.0 only computes one smoothed DEM.
Once the smoothed DEM is calculated, the slope aspect (0-360°) is computed and facets are defined. There are five (5)
facets in TIERv1.0: 1) North (aspect > 315°, aspect ≤ 45°); 2) East (45° < aspect ≤ 135°); 3) South (135° < aspect ≤ 225°);
4) West (225° < aspect ≤ 315°); and 5) Flat (D94). Flat aspects are areas with terrain gradients (slopes) less than the user
specified minGradient (m km⁻¹, Table 2). After the facets are defined, small facets are merged together with neighboring

facets using the minimum size model parameters (Table 2). Flat regions that are very narrow are considered ridges and behave like neighboring facets. These are merged into the neighboring facets on the west or south slopes depending on their orientation (D94).

### 2.2.2 Distance to the coast

The user defines a land /ocean mask field in the input grid file. This mask defines which grid points are large bodies of water (ocean points) and are used in the coastal proximity calculation. The distance to the coast is computed using the great circle distance assuming a spherical earth for every grid cell within a user defined distance threshold.

### 2.2.3 Topographic Position

To identify the atmospheric layer (Section 2.2.4) of a grid point the local topographic position of a grid point is computed

first. The topographic position calculation uses the high-resolution DEM. Following D02, for each grid point:

1) the minimum elevation within a user defined local search radius ($r$, Table 2) is found. D02 suggest a search radius of 40 km.

$$E_{M_{i,j}} = \min\big(E(i - r: i + r, j - r: j + r)\big) \tag{2}$$

where $E(i - r: i + r, j - r: j + r)$ denotes the DEM elevations within $\pm r$ grid points at valid land points in the $i$

and $j$ directions and $E_{M_{i,j}}$ is the local minimum elevation.

2) the topographic position is then estimated as:

$$T_{i,j} = E_{i,j} - \frac{(E_M(i - r: i + r, j - r: j + r))}{N_g} \tag{3}$$

where $N_g$ is the number of land grid points within the search radius ($N_g \leq r^2$).

### 2.2.4 Two-layer atmosphere

Following the determination of the topographic position each grid cell is placed into the first (boundary or inversion layer) or second layer (free atmosphere) of the idealized two-layer atmosphere. The height of the inversion layer is defined by the user (Table 2) and added to the mean elevation computed on the left-hand-side of Eq. (3). This defines an inversion height above sea level for all grid points. All grid points where $E_{i,j}$ less than the inversion height are placed into layer 1, while all other grid points are placed into layer 2 (D02).

### 2.2.5 Station metadata

After the input domain grid file has been processed, the preprocessing routine generates station metadata files for all precipitation and temperature stations that will be used in the regression model. Each station is assigned the closest grid point value of the smoothed DEM, facet, topographic position, atmospheric layer, and coastal distance.

### 2.3 Interpolation model

The regression model is applied to each land masked grid cell. It consists of routines to compute the station weights, to estimate the meteorological-terrain relationships, and to estimate the variable value at each grid point. A parameter and control file specify model parameters, and IO directories and files; see Tables 3 and 4 respectively. A flow chart describing the general flow, order of operations, and data requirements is given in Figure 2. Figures 3 and 4 provide more detailed flow charts for the specific processing flow for precipitation and temperature variables.

### 2.3.1 Station selection and weighting

For each grid point a set of stations are used to estimate the precipitation and temperature values. First, all stations within the user defined search radius are found (nearby stations), up to the maximum number of stations considered (Table 4). From that subset of stations, all stations on the same facet as the current grid point are identified (facet stations). Then a set of distance dependent weights and weights for each physical process component described in sections 2.2.1-2.2.4 are generated for all nearby and facet stations for each grid point. These component weights are then combined to create the final station weight vector.

$$W = W_d W_f W_l W_t W_p \tag{4}$$

where $W$ is the final weight vector, $W_d$ are the distance dependent weights, $W_f$ are the facet weights, $W_l$ are the atmospheric layer weights, $W_t$ are the topographic position weights, and $W_p$ are the coastal proximity weights. All component weights and the final weight vector are normalized to sum to unity (D02). For precipitation, only $W_d$, $W_f$, and $W_p$ are used to estimate $W$, while temperature uses all five component weights in $W$.

### 2.3.1.1 Distance dependent weighting

A stations relevance to the current grid point decreases as the station distance increases (e.g. Shepard 1968), thus this component station weighting decreases with increasing distance. Here we generally follow the synagraphic computer mapping (SYMAP) algorithm of Shepard (1968, 1984) and develop inverse distance weights that are further modified by including direction information. Direction information is used to downweigh stations that are at a similar direction but further distance as other stations, as their influence has been 'shadowed' by the nearer station (Shepard 1968). The distance weighting function of Barnes (1964) is used:

$$I = \exp\left(-\frac{d^y}{s}\right) \tag{5}$$

where $I$, are the inverse distance dependent weights, $d$ is the station distance vector, and $y$ and $s$ are the user defined Barnes exponent and scale factor respectively (Table 4). The angle dependent weights are then computed as (Shepard 1984):

$$T_s = \sum_{s \neq q} I_q \left(1 - \cos(A_s - A_q)\right) \tag{6}$$

where $T_s$ is the station angle weight for station $s$ and subscripts $s$ and $q$ denote station subscripts for stations $1{:}n_x$ where $n_x$ is the maximum number of stations considered for each grid point (Table 4). The final distance dependent weights are then computed as:

$$W_{d,s} = I_s^2 \left(1 + \frac{T_s}{\sum_{s=1}^{n} \sum T_s}\right) \tag{7}$$

where $n$ is the number of stations considered at the current grid point. $W_d$ is then normalized to sum to unity.

**2.3.1.2 Facet weighting**

Stations on the same facet type as the current grid point receive an initial facet weight of 1. D02 introduces a method to reduce the weight of stations on the same facet type but with intervening facets of different types between the station and grid point (D02 equation 5). This is not implemented here and all stations on the same facet type as the current grid point receive the same weight. This could be a decision considered for exploration in a future TIER release. The distance

dependent weights already account for this implicitly, but the explicit inclusion of additional weight decreases for stations on the same facet type will increase the localization of the TIER station weighting even further. This would increase the small scale features of TIER.

**2.3.1.3 Atmospheric layer**

The atmospheric layer weight function is defined as:

$$W_{l,s} = \begin{cases} 1, & \Delta l_s = 0 \\ \dfrac{1}{(E_{i,j}-E_s)^a}, & \Delta l_s = 1 \end{cases} \tag{8}$$

where $\Delta l_s$ is the layer difference between the grid point and station $s$, the elevations are defined using the high-resolution DEM and station elevation ($E_s$), and $a$ is the user defined layer weighting exponent (Table 4). Following D02, stations in the same atmospheric layer as the current grid point receive an initial weight of 1. D02 includes an additional check to see if the station – grid elevation difference is smaller than some threshold value for a station. If this is true a station in a different

layer and grid cell may still receive a weight of 1. TIERv1.0 does not include the additional conditional statement and only stations in the same atmospheric layer receive an initial weight of 1. The vertical elevation difference is then used to weigh the remaining stations. The atmospheric layer weighting is applied only to temperature variables in TIERv1.0.

### 2.3.1.4 Topographic position

The topographic position weights are computed following D07:

$$W_{t,s} = \begin{cases} 1, & \Delta t_s \leq \Delta t_m \\ 0, & \Delta t_s > \Delta t_x \\ \frac{1}{\Delta t_s^z}, & \Delta t_m < \Delta t_s \leq \Delta t_x \end{cases} \tag{9}$$

where $\Delta t_s$ is the topographic position difference between the current grid cell and station $s$, $\Delta t_m$ and $\Delta t_x$ are the user defined minimum and maximum topographic position differences, and $z$ is the topographic position weighting exponent (Table 4). The topographic position weight enhances identification of stations that lie in similar topographic areas (e.g. valleys) and is applied only to temperature variables in TIERv1.0.

### 2.3.1.5 Coastal proximity

Using the computed distance to the coast, the coastal proximity weights are computed as:

$$W_{p,s} = \begin{cases} 1, & \Delta p_s \leq 1 \\ \frac{1}{(\Delta p_s)^c}, & \Delta p_s > 1 \end{cases} \tag{10}$$

where $\Delta p_s$ is the absolute difference between the current grid cell and station $s$ distance to the coast values, and $c$ is the user defined coastal proximity weighting exponent (Table 4). D02 computes coastal proximity weights using the same inverse distance function, but also includes a threshold, $\Delta p_x$ where if $\Delta p_s > \Delta p_x$, is set to zero. This weighting factor highlights stations with similar coastal proximity to the current grid cell.

### 2.3.2 Grid point estimate

Once nearby stations are selected and the final weight vector is computed (Eq. (4)), a base grid point estimate is developed using the weighted average of all nearby stations:

$$\hat{\mu}_{b_o} = \sum_{s=1}^{n} Y_s * W_s \tag{11}$$

where $\hat{\mu}_{b_o}$ is the grid point meteorological variable estimate, and $Y_s$ and $W_s$ are the observed station value and the station weight for station $s$ respectively. The uncertainty of this value is estimated as the standard deviation of the leave-one-out estimates, which is all possible combinations of $n_r$-1 stations, $\binom{n_r}{n_r-1}$, which in this case are $n_r$ possible combinations.

$$\hat{\sigma}_{b_o} = \sqrt{\frac{\sum_{i=1}^{n_r} \left(\hat{\mu}_{b,-1}\right)^2}{n_r - 1}} \tag{12}$$

where $n_r$ is the subset of stations that are both within the distance threshold and on the same facet as the current grid cell, $\hat{\sigma}_{b_o}$ is the estimated standard deviation of $\hat{\mu}_{b_o}$, and $\hat{\mu}_{b,-1}$ is the estimated value when the $i$-th station is withheld.

Next, the variable – elevation linear regression coefficients are solved for:

$$\hat{\boldsymbol{\beta}} = (\mathbf{A}_s^\mathrm{T} \mathbf{W} \mathbf{A}_s)^{-1} \mathbf{A}_s^\mathrm{T} \mathbf{W} \boldsymbol{Y} \tag{13}$$

where $\boldsymbol{\beta}$ is the vector of linear regression coefficients $(\beta_o, \beta_1)$, $\mathbf{A}_s$ is the $n_r$ x 2 design matrix, $\mathbf{W}$ is the $n_r$ x $n_r$ diagonal weight matrix populated with $W$, and $\boldsymbol{Y}$ is the vector of observed station values. In D94, D02, and D08 these coefficients

determine the grid point estimate as:

$$\hat{\mu} = \hat{\beta}_0 + \hat{\beta}_1 E_{i,j}, \ \ \beta_{1M} \leq \hat{\beta}_1 \leq \beta_{1X} \tag{14}$$

where $\beta_{1M}$ and $\beta_{1X}$ are the user defined minimum and maximum valid regression slopes (Table 4). Note that slope here is in physical units per distance (e.g. mm km$^{-1}$ or K km$^{-1}$), which is also referred to as the lapse rate in atmospheric science. In TIERv1.0 we have chosen to use the base grid point estimate, $\hat{\mu}_b$ as the intercept value in the variable-elevation regression

equation. This is done because when $\hat{\beta}_1$ falls outside of the bounds in Eq. (14), a default slope value is used, but $\hat{\beta}_0$ is not modified. Thus, the base estimate of a variable for a grid cell is sometimes derived from an equation the system considers invalid. Therefore, we fully disassociate the intercept and slope estimates. Here we provide an initial assessment of this choice in section 3, but this methodological choice should be examined in more detail in future work. Subsequently we then also modify the elevation used in the regression equation to be the difference between the high resolution DEM elevation

and the $W$ weighted station elevation using the smoothed DEM station elevations. The switch to an elevation difference is required as we are effectively correcting the base estimate to the DEM elevation, and the base estimate has an intrinsic elevation associated with it. Therefore the TIERv1.0 grid point estimate is:

$$\hat{\mu} = \hat{\mu}_{b_o} + \hat{\beta}_1 \Delta E, \ \ \beta_{1M} \leq \hat{\beta}_1 \leq \beta_{1X} \tag{15}$$

where $\Delta E$ is the difference between the smoothed DEM elevation and the $W$ weighted station elevation using the smoothed

DEM station elevations. When $\hat{\beta}_1$ is invalid the default slope is used, and when the initial $\hat{\beta}_1$ is valid, the uncertainty of $\hat{\beta}_1$ is estimated in a similar manner to that of $\hat{\mu}_{b_o}$ using Eq. (12). Note that for temperature variables only, the user can define a spatially variable default lapse rate (Table 3, Figure 4). The standard deviation of all valid slope estimates from the leave-one-out estimates, $\binom{n_r}{n_r - 1}$, is used as the uncertainty estimate of $\hat{\beta}_1$:

$$\hat{\sigma}_{\beta_1} = \sqrt{\frac{\sum_{i=1}^{n_r} (\hat{\beta}_{1,-1})^2}{n_r - 1}} \tag{16}$$

where $\hat{\sigma}_{\beta_1}$ is the estimated standard deviation of $\hat{\beta}_1$ and $\hat{\beta}_{1,-1}$ is the estimated value when the $i$-th station is withheld.

D02 define the method to adaptively adjust the station search radius until the minimum number of needed stations is met. Here we do not adjust the search radius and instead attempt the regression and uncertainty estimation when $n_r \geq n_m$, where $n_m$ is the user specified minimum number of stations required for the regression (Table 4). When $n_r < n_m$, the regression is

attempted for $2 \leq n_r < n_m$ and the default slope is used when $n_r < 2$. Additionally, for $n_r < n_m$, Eq. (16) is never applied and there is no direct uncertainty estimate of $\hat{\beta}_1$ for those grid cells.

Finally, D94 found that normalizing the precipitation lapse rate (km$^{-1}$) after performing the regression reduces the large spatial variability in precipitation lapse rates due to the large spatial variability in the underlying precipitation amounts. The normalization is done at each grid cell as:

$$\widehat{B}_{1P} = \frac{\beta_{1P}}{\overline{Y_P}}$$ (17)

where

$$\overline{Y_P} = \frac{1}{n_r}\sum_{s=1}^{n_r} Y_{P,s}$$

and $\overline{Y_P}$ is the mean precipitation (mm) of all stations considered for the regression for the current grid point, $\widehat{\beta}_{1P}$ is the estimated slope in physical units (mm km$^{-1}$), and $Y_{P,s}$ is the station precipitation (mm) at station $s$. Accordingly, $\widehat{\sigma}_{B_{1P}}$ is:

$$\widehat{\sigma}_{B_{1P}} = \sqrt{\frac{\sum_{i=1}^{n_r}(\widehat{B}_{1P,-1})^2}{n_r-1}}$$ (18)

The normalization allows for the bounds in Eqs. (14-15) to be broadly applicable for precipitation, as well as for a reasonable default lapse rate to be applied to grid points where a valid regression slope cannot be found. Temperature lapse rates are computed in physical units (K km$^{-1}$) as there is little variability in temperature lapse rates.

## 2.4 Post-processing

Several post-processing steps are undertaken to reach the final gridded estimates after all grid points have an initial estimate, shown in Figure 5. These include updating estimated slope values, applying spatial filters, and recomputing the final fields.

### 2.4.1 Precipitation

The initial precipitation normalized slope estimates are used to recompute the default slope if the user specifies (Table 4). In this case, all grid points with valid regression slopes are used to compute the domain mean normalized precipitation slope. This value is then substituted at all grid points with default slope estimates. Next, a 2-D Gaussian filter is applied to the normalized slopes to reduce noise and smooth the artificial numerical boundaries in slope values and is taken as the final precipitation slope estimate (Fig. 5a). The parameters of this spatial filter (size and spread) are specified in the TIER model parameter file (Table 4).

After the slope estimates have been finalized, the precipitation is field is recomputed using Eq. (15) and then a feathering process is applied to smooth any remaining very large gradients (e.g. D94, Fig. 5a). The feathering routine operates on the normalized precipitation slopes and searches for grid cell to grid cell gradients in the normalized slope larger than a user specified value (Table 4). If a large gradient is found, the slope of the grid cell with less precipitation is increased until the gradient falls below the maximum allowable value. The feathering routine iterates over the grid until there are no remaining

large gradients and is an additional smoothing step for precipitation in TIER. Also, the feathering routine only runs for grid cells with larger elevation changes than a user specified minimum gradient (Table 4), which effectively ignores flat areas (D94), and in TIERv1.0 the feathering routine only operates on grid cells above a user specified minimum elevation.

Finally, uncertainty estimates are recomputed for the entire grid, first for the base estimate and slope components, then for the total uncertainty of Eq. (15) (Fig. 5a). For those grid points with no initial $\hat{\sigma}_{B_1}$, the nearest neighbor estimate is used. Then the same Gaussian filter applied to the normalized precipitation slopes is applied to the gridded $\hat{\sigma}_{b_o}$ and $\hat{\sigma}_{B_1}$. The final uncertainty contribution due to uncertainty in the precipitation slope at a grid point in physical units (mm) is then computed as:

$$\hat{\sigma}_{\beta P} = \hat{\sigma}_{\beta_1 P}\,\hat{\mu}_P\,\text{abs}(\Delta E) \tag{19}$$

where $\hat{\sigma}_{\beta P}$, is the final uncertainty (mm) due to uncertainty in the precipitation slope, $\hat{\sigma}_{\beta_1 P}$ is the final slope uncertainty (mm km$^{-1}$), and $\hat{\mu}_P$ is the final precipitation estimate (mm). The filtered $\hat{\sigma}_{b_o P}$ field is used as the final base precipitation estimate, $\hat{\sigma}_{bP}$. The total uncertainty is estimated as the combined standard deviation of the two component estimates:

$$\hat{\sigma}_P = \hat{\sigma}_{bP} + \hat{\sigma}_{\beta P} + 2\sqrt{\text{cov}(\hat{\sigma}_{bP}, \hat{\sigma}_{\beta P})} \tag{20}$$

because the covariance between the two component uncertainties is sometimes nonzero. The covariance is computed locally at each grid point using a user defined 2-D window of points (Table 4) around the current grid point.

**2.4.2 Temperature**

Post-processing for temperature is simpler than for precipitation because the temperature lapse rates are in physical units. The initial valid temperature slope estimates are used to recompute the default lapse rate if the user specifies (Table 4) when
there is no spatially varying default temperature lapse rate information provided (Table 3). Again, the mean of all valid regression slope estimates is used as the updated default temperature lapse rate for this case. As for precipitation, a 2-D Gaussian filter is then applied to the slopes to reduce noise and smooth the artificial numerical boundaries in slope values and is taken as the final temperature slope estimate (Fig. 5b). Then the final temperature estimate is computed using these updated lapse rate values and Eq. (15).

As for precipitation, the component and total uncertainty estimates are then finalized for temperature. The base temperature estimate uncertainty and slope uncertainty are smoothed using the 2-D Gaussian filter to estimate the final component uncertainties, $\hat{\sigma}_{bT}$ and $\hat{\sigma}_{\beta_1 T}$ respectively. Then the final temperature uncertainty contribution due to temperature lapse rate uncertainty is computed using Eq. (18), and Eq. (19) is used to compute the total uncertainty of the temperature estimate,
substituting subscript Ts for Ps in both.

## 3 Model evaluation and sensitivity experiments

An example use case over the western United States, focused primarily on the Sierra Nevada mountains between roughly 35 °N to 43 °N and 118 °W to 125 °W including precipitation, maximum ($T_{max}$) and minimum ($T_{min}$) temperature data (Fig. 6) is used for computing basic model evaluation statistics. This evaluation is to simultaneously determine if the TIERv1.0 algorithm is performing as expected numerically and to provide a brief baseline of performance. We calculate bias and mean absolute error (MAE) statistics from the final gridded meteorological variables using all available stations, or a calibration sample evaluation. Additional evaluation is considered outside the scope of this initial presentation of the model.

The gridded output fields are nearly unbiased for all three variables, 0.2 mm, -0.22 K and -0.21 K for precipitation, $T_{max}$, and $T_{min}$ respectively. MAE values are 0.84 K $T_{max}$ and 0.75 K $T_{min}$ and 14.3 mm for precipitation (Table 5). Additionally, the gridded output have nearly zero conditional bias for temperature as indicated in Figure 7a-b, where the fitted slope to the TIER-observation points is 0.93 and 0.96 for $T_{max}$ and $T_{min}$ respectively. There is an overestimation at smaller values transitioning to an underestimation at larger values. Precipitation has the same conditional bias structure as temperature (Fig. 7c), however the slope of the TIER-observation fitted linear regression is 0.88, indicating a larger conditional bias as observed precipitation increases.

Figure 8 highlights the methodological choice made in section 2.3.2 to disassociate the intercept parameter from the regression estimated slope in Eq. (15) for precipitation. We compare estimates using $\hat{\beta}_0$ (PRISM-similar) in Eq. (15) versus $\hat{\mu}_{b_o}$ (TIER v1.0) and find that in general precipitation estimates using $\hat{\beta}_0$ are larger than that of TIER v1.0, particularly at higher elevations. TIERv1.0 has mean precipitation for grid points below and above 2000 m of 83.2 and 88.6 mm respectively. The PRISM-similar method and has average precipitation values of 117.6 and 152.3 mm above and below 2000 m, which are 42% and 72% increases over TIER v1.0. Comparing to in-sample station observations shows that the $\hat{\beta}_0$ estimation method results in higher biases and MAE than TIER v1.0; 38.2 vs 0.2 mm bias and 51.9 vs 14.3 mm MAE for the two methods respectively.

These differences could be due to several reasons including that TIER v1.0 parameters were subjectively tuned for the published methodology. Also, in-sample validation does not truly determine method performance, an out of sample verification exercise and further evaluations should be undertaken. The PRISM-similar method within the PRISM model performs extremely well and may likely be more appropriate for higher elevations given the tendency for these types of linear regression systems to underestimate precipitation above the highest observation when using smoothed DEM values (see section 4d.4 and parameter B1EX in Table 1 of D94).

### 3.1 Model Parameter Experiments

Here we explore the impact of model parameter changes on the output values and their associated uncertainty estimates. We modify TIER model parameters only (no preprocessing parameters) and make three parameter changes to parameters focused on different parts of the interpolation model for different variables in an effort to concisely highlight how model parameter choices impact the final product. First, we modify the inverse distance weighting exponent in the distance dependent weighting function for $T_{min}$ (experiment 1), then we modify the coastal distance weighting exponent for precipitation (experiment 2), and finally we modify the maximum number of stations allowed for each grid point for precipitation (experiment 3).

### 3.1.1 Experiment 1

In Experiment 1 the parameter 'distanceWeightExp' (Table 4), which is the exponent in the distance dependent weighting function (Eq. (5)), is modified from 2 (default), to 1.75 (modified) for a spatial simulation of $T_{min}$. This decreases the negative slope of the inverse distance weighting function such that stations further from the considered grid point receive more weight in the modified case than the default. The resulting $T_{min}$ distributions and difference field are given in Figure 9. The spatial distributions are very similar throughout most of the domain (Fig. 9) as the observation network is relatively high density across most of the domain (Fig. 6). Where the station density decreases along the eastern side of the domain, differences increase in magnitude east of 119 °W. Notably there are also pockets of differences outside of ± 1 °C in areas with high station density along the coast and between 40-42 °N, 121-123 °W (Fig. 9c). These locations contain complex terrain, specifically large elevation gradients, and the modified station weights result in different estimated temperature lapse rates in addition to changes in the base estimate resulting in the different temperature estimates. However, the calibration sample statistics are not significantly different at the 90% confidence level than the default parameter set (Table 6), suggesting that the changes in the gridded field are not able to be differentiated in a meaningful way.

### 3.1.2 Experiment 2

For experiment 2 we examine precipitation and modify the 'coastalExp' (Table 4) parameter from 0.75 to 1 to examine the influence of changes to the coastal proximity weighting. Qualitatively, the two precipitation distributions are identical with the overall precipitation pattern remaining essentially unchanged (Figure 10a-b). The difference fields show that there are shifts in the precipitation placement throughout the domain through the alternating positive/negative difference patterns, particularly across the complex terrain (Fig. 10b), but essentially no net precipitation change with a total relative difference of 0.2% between the two estimates. Absolute differences can be as large as 46 mm in areas of large total accumulations, however the relative differences in those areas are generally less than 10% (Fig. 10b). Correspondingly, dry areas have smaller absolute differences, but sometimes larger relative differences, as can be seen along the eastern third of the domain

(Fig. 10b). The mean absolute value of the cell to cell precipitation gradient is 1.56 mm km$^{-1}$ versus 1.69 mm km$^{-1}$ (8.3% increase) in the base and modified cases respectively. Increased station localization should be expected to increase high-frequency variability and thus spatial gradients. The calibration sample statistics are not significantly different at the 90% confidence level from the default parameter set (Table 6). However in this case the confidence bounds are almost all non-overlapping, which may suggest increasing the coastal exponent further would improve the model performance.

The total uncertainty is generally increased by a few mm across the domain (0.55 mm on average), with a corresponding relative increase in uncertainty of around 5-10% (Figure 10c). This is due to the fact that increasing this weight exponent decreases the weight of stations more dissimilar to the current grid point, effectively increasing the localization of the weights and increasing the variability of the leave-one out estimates (Eq. (16)).

### 3.1.3 Experiment 3

Finally, we change the 'nMaxNear' parameter (Table 4) from 10 to 13, which controls the maximum number of stations used for each grid point the interpolation model. Again, the precipitation pattern is essentially qualitatively unchanged between the two configurations with a 0.2% domain average change, also see Figure 11a. However, the relative difference fields highlight larger and more systematic changes to the precipitation distribution than in experiment 2. Areas of the highest accumulation in the base case have less precipitation in the modified case (compare Fig. 10a and Fig. 11a) with 89% (217/245) of the grid points having precipitation > 300 mm in the base case having less precipitation in the modified case. Conversely, 57% (7686/13430) of the grid points having < 300 mm in the base case have more precipitation in the modified case. This is because generally more stations are included in the estimate for each grid cell, which results in smoother final estimate through smoothed base and slope precipitation estimates in Eq. (15). The mean absolute value of the cell to cell precipitation gradient is 1.56 mm km$^{-1}$ versus 1.48 mm km$^{-1}$ (5.5% decrease) in the base and modified cases respectively. The calibration sample statistics are statistically equivalent to the base case, but the MAE in experiment 3 is statistically significantly larger than experiment 2. This is an expected result given that the final estimate is less localized for any specific station.

Increasing the number of stations considered reduces the estimated uncertainty across nearly the entire domain (Fig. 11b-c). On average there is a 2.1 mm (16%) reduction in the domain mean uncertainty with some grid cells having reductions > 25 %. The large decreases are primarily in regions of complex terrain, and this is controlled by changes in the slope uncertainty estimate, $\hat{\sigma}_{\beta P}$ (Fig. 11c). This change in total uncertainty is slightly larger, but opposite in sign to the parameter modification in experiment 2.

## 4 Summary and Discussion

The Topographically InformEd Regression (TIER) software was developed for several reasons. First, the systems for spatial modeling of meteorological variables from *in situ* observations have matured to the point that they are complex systems with many methodological choices and model parameters. TIERv1.0 provides an initial implementation of a knowledge-based

statistical modeling system based on D94, D02, D00, D07, and D08 with the capability to explore different methodological choices in a systematic fashion. The system is modular so that new knowledge-based ideas can be added to the regression model through including new weighting terms. Model parameters are also accessible to the user allowing for parameter perturbation experiments. More broadly, this should be viewed as a first step towards development of flexible, open-source systems that include many of the commonly used spatial interpolation models so the community can more fully understand

methodological choices in gridded meteorological product generation (e.g. Newman et al. 2019). Understanding how methods and model parameters interact and modify the final output is key to improving these systems.

The parameter experiments performed here provide three examples highlighting how minor changes to one model parameter impact the final spatial distribution. For example, modifying the coastal weight exponent results in a shift in placement of

precipitation across the domain (Fig. 10) and systematic changes in the estimated uncertainty. Increasing the maximum number of stations considered for the interpolation results in systematic changes to the precipitation distribution and decreases the sharpness of the final field (Fig. 11). Also, the spatial gradients of precipitation and total uncertainty changes are of opposite sign for experiments 2 and 3. In general, parameter changes that act to increase localization will enhance gradients and uncertainty, while those that decrease localization or increase sample sizes will decrease gradients and

uncertainty. This highlights that parameter interactions could play a role in the final result through positive or negative feedbacks. Finally, experiments 2 and 3 result in non-significant calibration validation results as compared to the base case, while the MAE between the modified parameter sets in experiments 2 and 3 results in statistically significant MAE differences.

Given the ability to perform parameter sensitivity experiments in TIER, we reemphasize the need for novel evaluation methods including out of sample station networks (e.g. Daly 2006; Daly et al. 2017; Newman et al. 2019) that are as independent from the input networks as possible and integrated validation methods using ancillary observations such as streamflow and other modeling tools such as hydrologic models (Beck et al. 2017; Henn et al. 2018; Laiti et al. 2018).

Finally, TIER does not implement the exact system developed by Daly and colleagues and will not produce the same climate fields even with the same input data. TIER is not duplicating source code and every feature described in D94, D00, D02, D07, and D08, as TIER was developed as a knowledge-based system following these papers, not replicating them and other unpublished details. Also, TIER version 1.0 does not contain station input data preprocessing routines. Instead, example

input data are provided in the example cases dataset (section 5).  Station preprocessing and quality control can encompass a vast number of methods (e.g. Serreze et al. 1999; Eischeid et al. 2000; Durre et al. 2008, 2010; Menne and Williams 2009). These methods may be included in future releases or as separate community station quality control tools.

## 5 Code availability

The TIERv1.0 code is available at https://doi.org/10.5281/zenodo.3234938.  The active development repository of TIER is located at https://github.com/NCAR/TIER.

## 6 Data availability

The input data for the example domain used here are available at: https://ral.ucar.edu/solutions/products/the-topographically-informed-regression-tier-model

## 7 Author Contribution

AJN and MPC developed the TIER model concept.  AJN implemented the model, developed the test case, model validation, and model sensitivity experiments.  AJN and MPC contributed to the manuscript.

## 8 Competing Interests

The authors declare that they have no conflict of interest.

## Acknowledgements

The US Army Corps of Engineers (USACE) Climate Preparedness and Resilience program funded this work.  Color tables used here are provided by Wikipedia (precipitation), GMT (difference plots), and the GRID-Arendal project (http://www.grida.no/) (temperature) via the NCAR NCL and cpt-city color table archives (https://www.ncl.ucar.edu/Document/Graphics/color_table_gallery.shtml, http://soliton.vm.bytemark.co.uk/pub/cpt-city/).

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

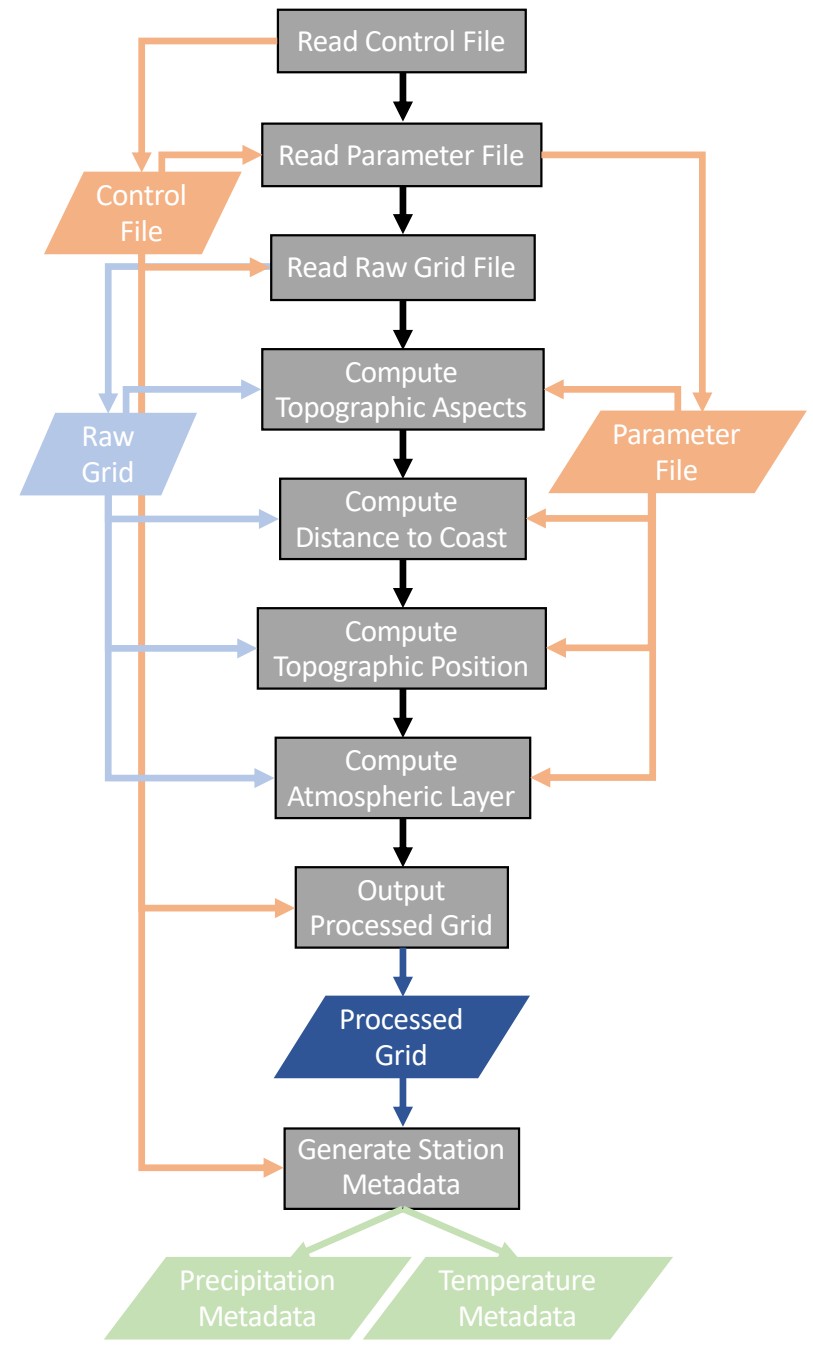

**Figure 1: Flow chart describing the TIER pre-processing system. Processes are shaded gray, input files are orange, topographic inputs and outputs are shades of blue, and outputs are various shades of green.**

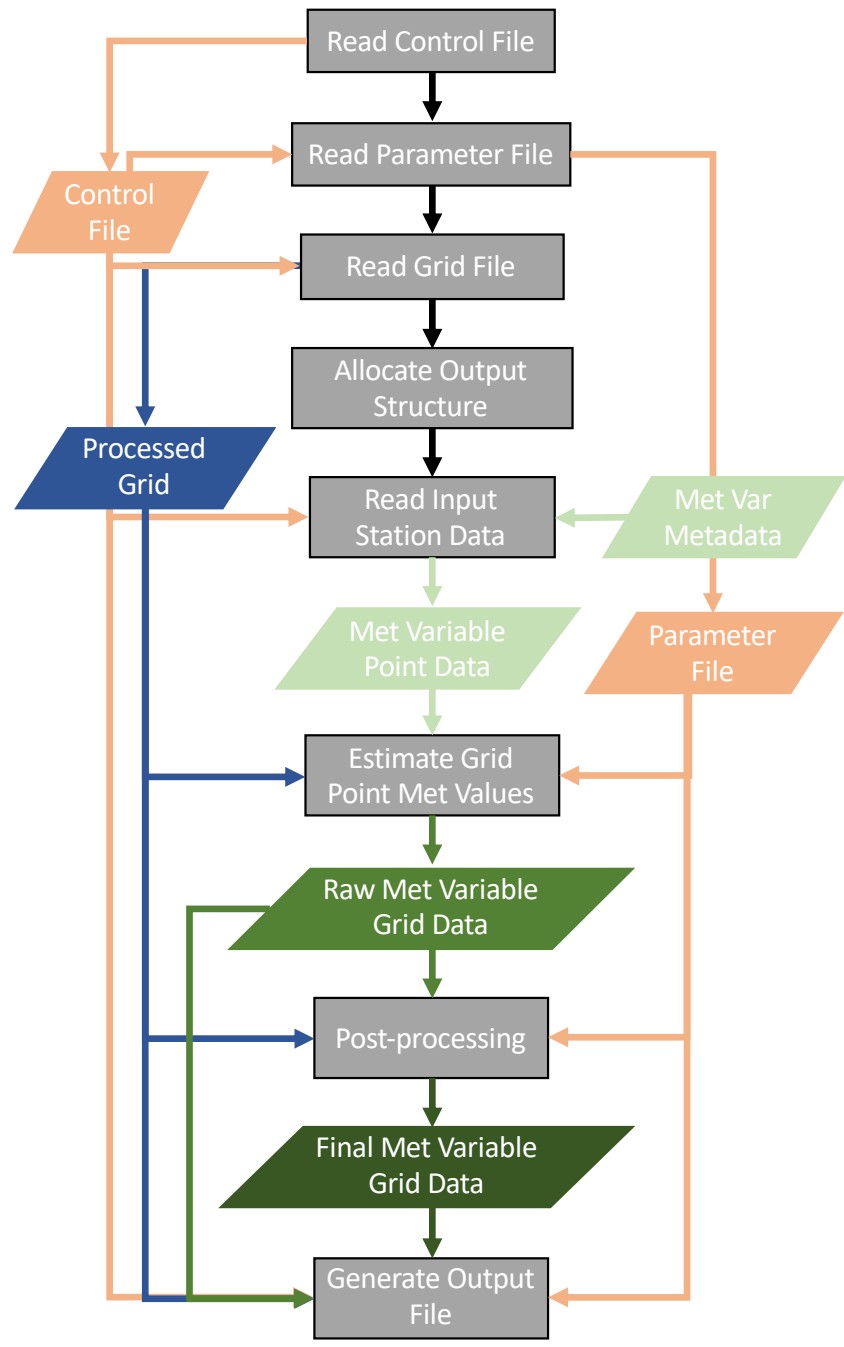

**Figure 2: Flow chart describing the TIER processing algorithm including post-processing. Color shading is the same as Figure 1.**

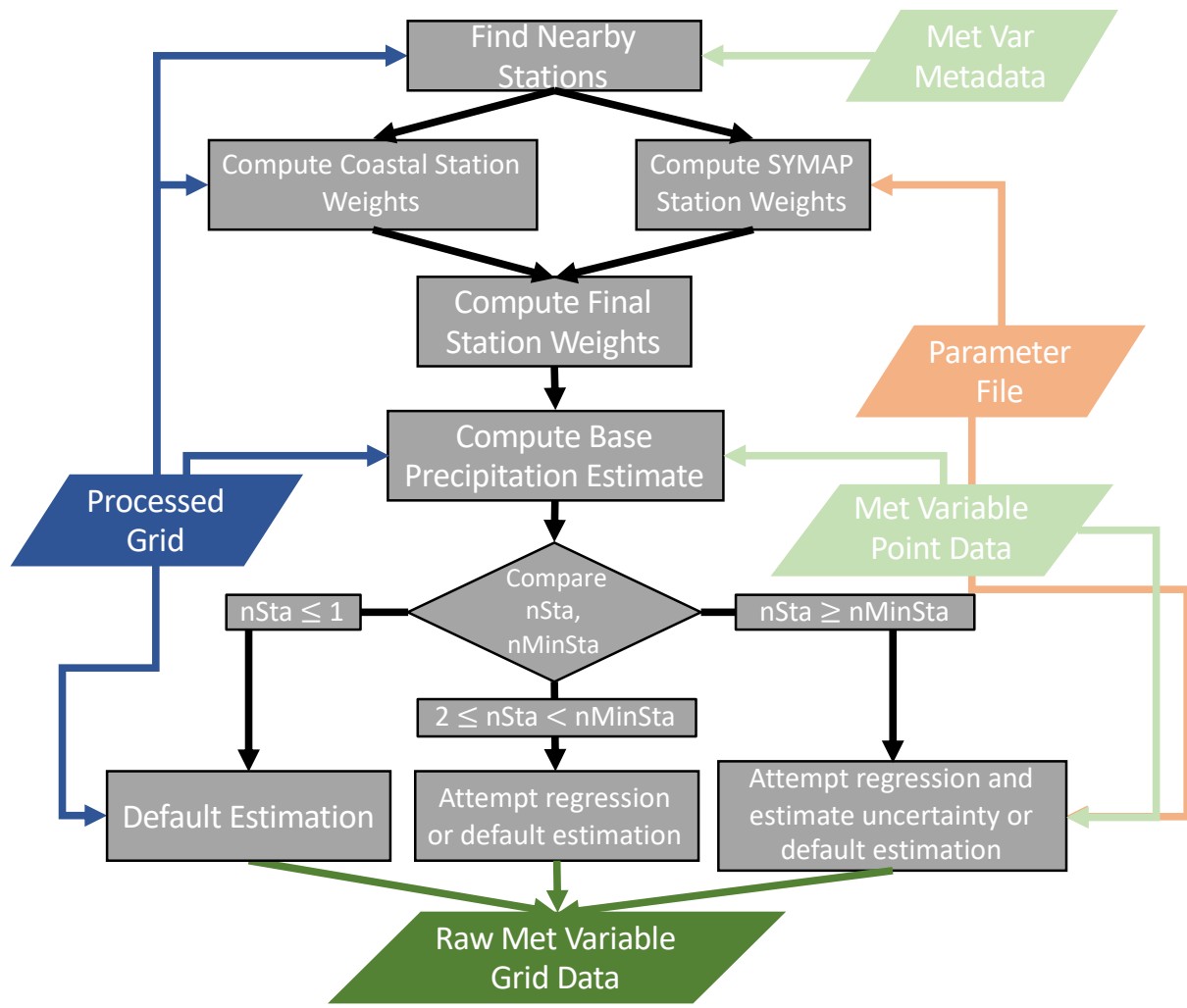

**Figure 3: Flow chart describing the precipitation grid point estimate algorithm. Color shading is the same as Figure 1.**

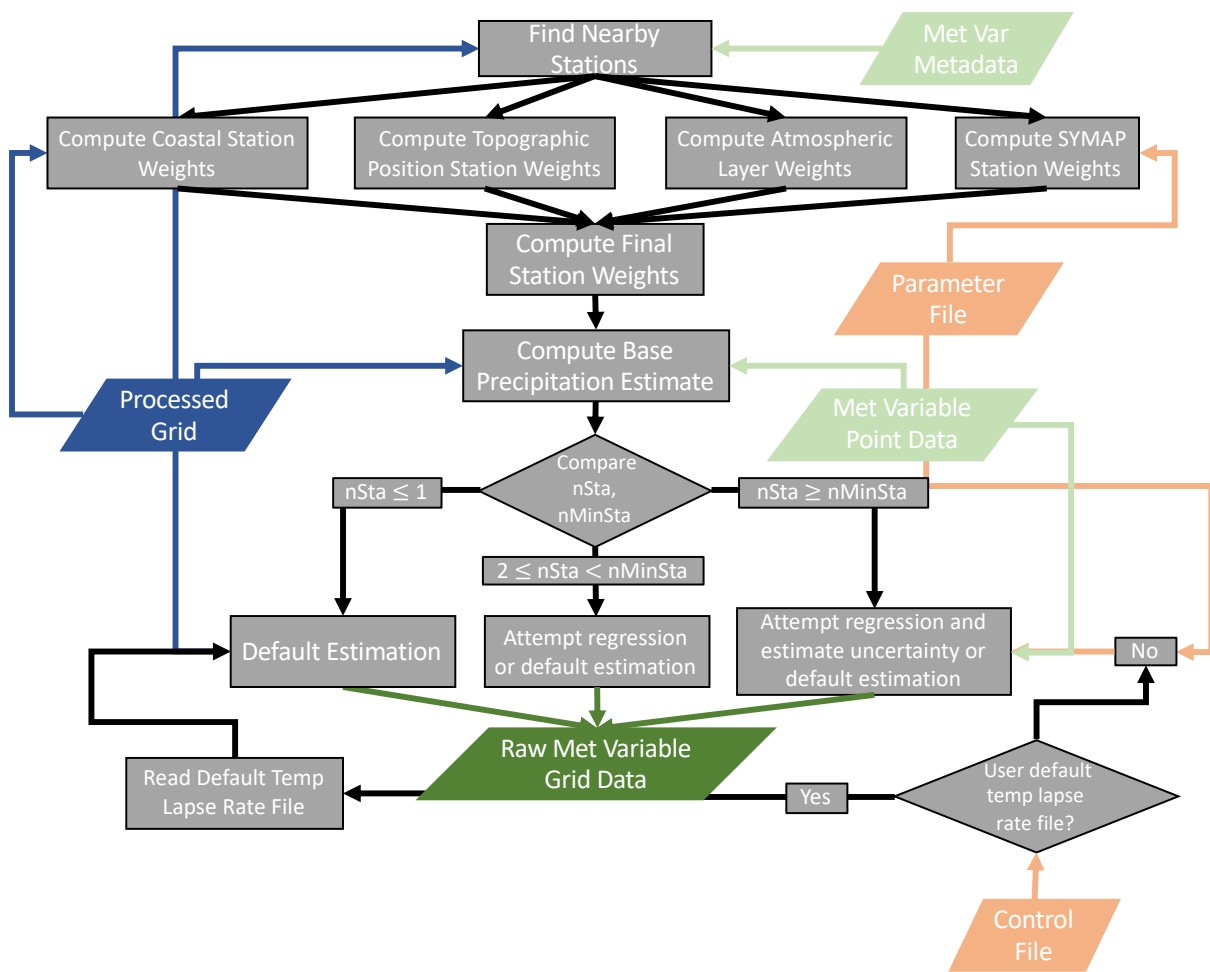

**Figure 4: Flow chart describing the temperature grid point estimate algorithm. Color shading is the same as Figure 1.**

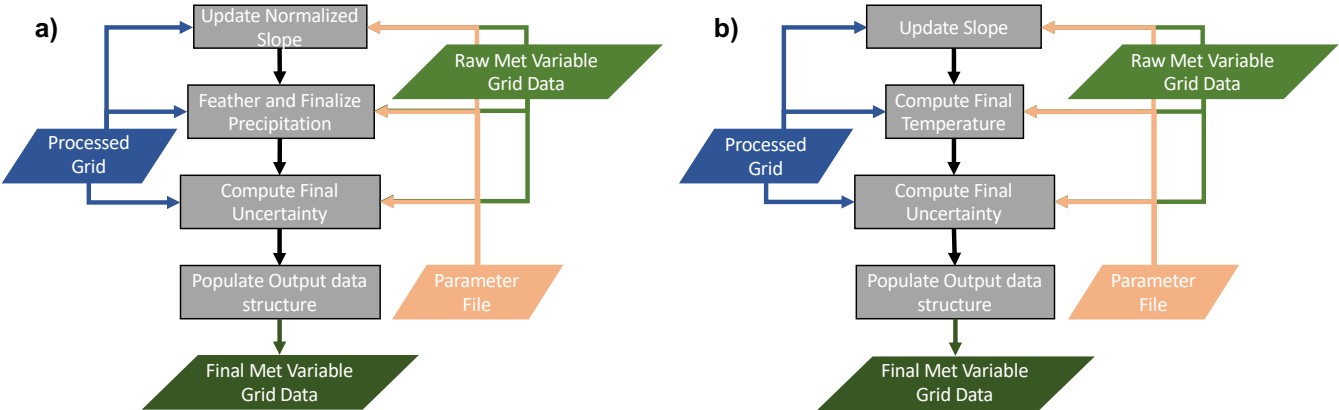

**Figure 5: Flow charts describing the a) precipitation post-processing, and b) temperature post-processing. Color shading is the same as Figure 1.**

## a) Temperature Observations

## b) Precipitation Observations

**Figure 6: The TIER test domain with a) the temperature station distribution, and b) the precipitation station distribution. Contours indicate the 0 m, 500 m, 1500 m, and 2500 m elevation contours moving from black to light gray.**

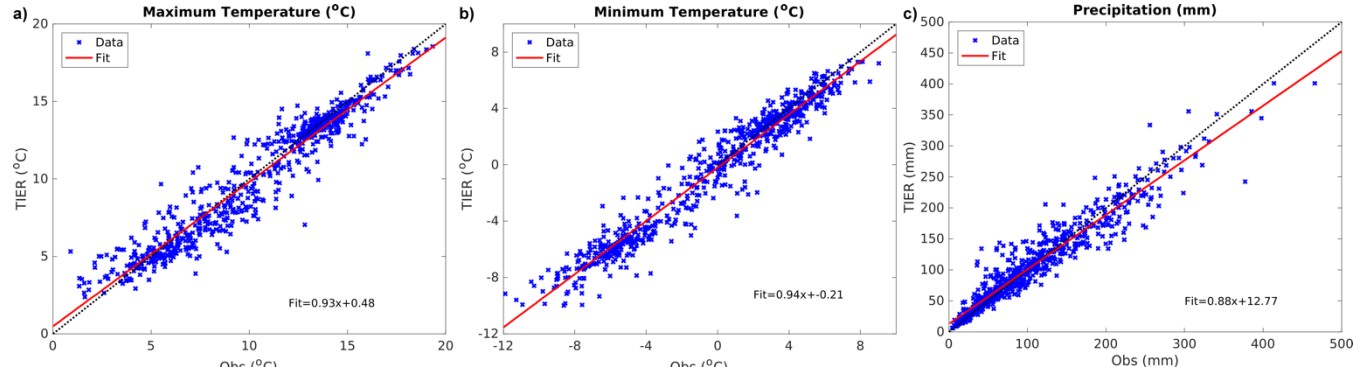

**Figure 7: Calibration sample evaluation scatter plots (TIER vs observations) for a) maximum temperature (°C), b) minimum temperature (°C), and c) precipitation (mm).**

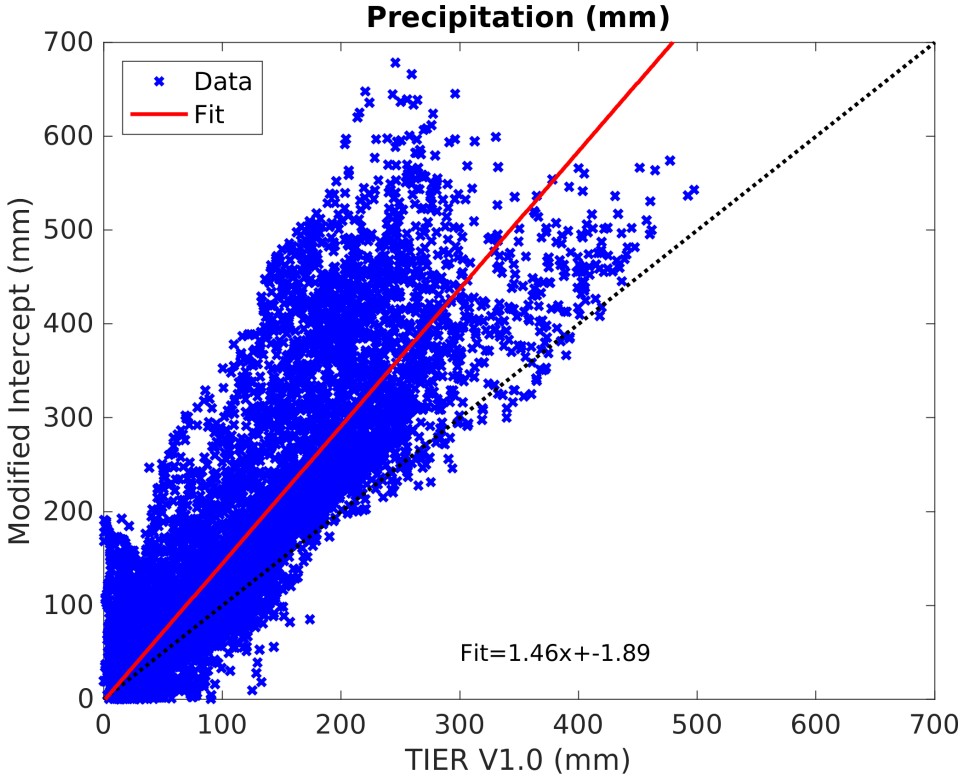

**Figure 8: Comparison of precipitation (mm) estimates using $\widehat{\beta}_0$ in Eq. 15. versus TIER v1.0 which uses $\widehat{\mu}_{b_o}$ in Eq. 15.**

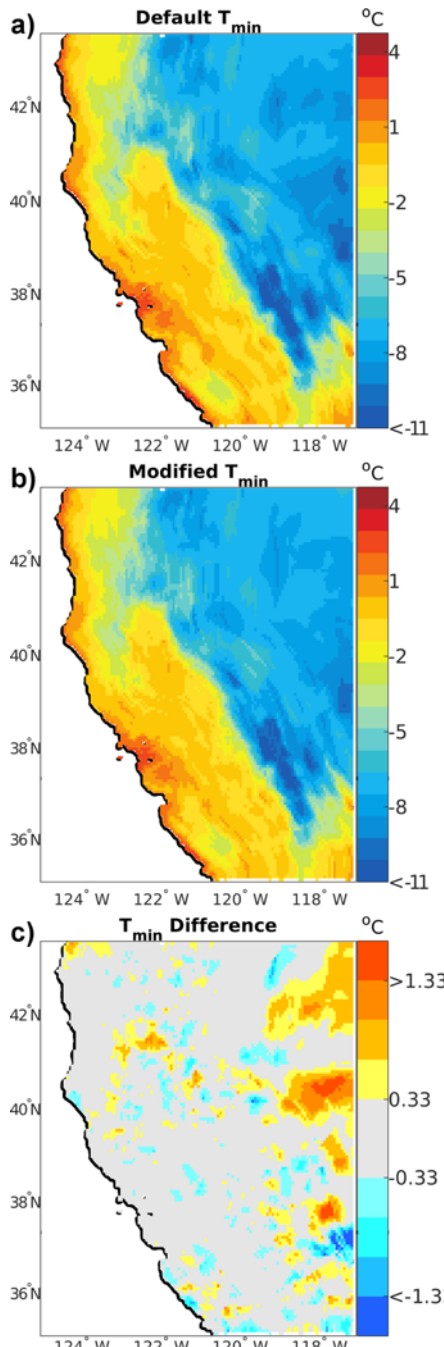

**Figure 9: Spatial distribution of minimum temperature (T$_{min}$, °C) for model parameter sensitivity experiment 1.  a) default model parameters, b) modified distance weighting exponent, and c) the difference field (default – modified).**

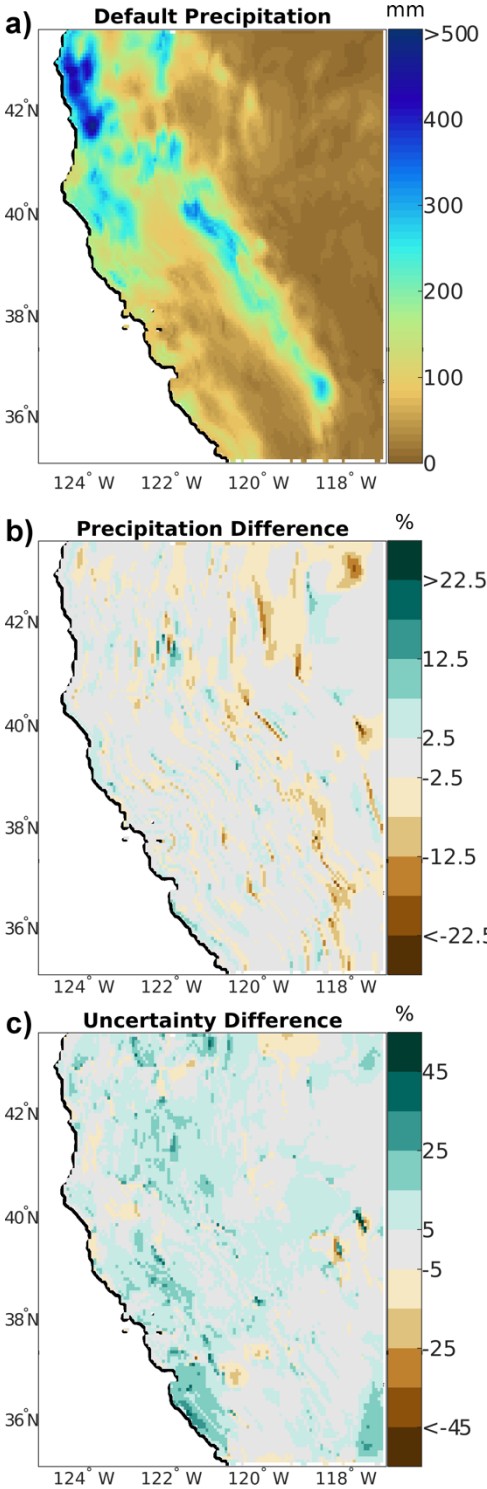

**Figure 10: Spatial distribution of a) base precipitation (mm) for model parameter sensitivity experiment 2, b) default – modified precipitation difference (mm), and c) default – modified uncertainty difference (%).**

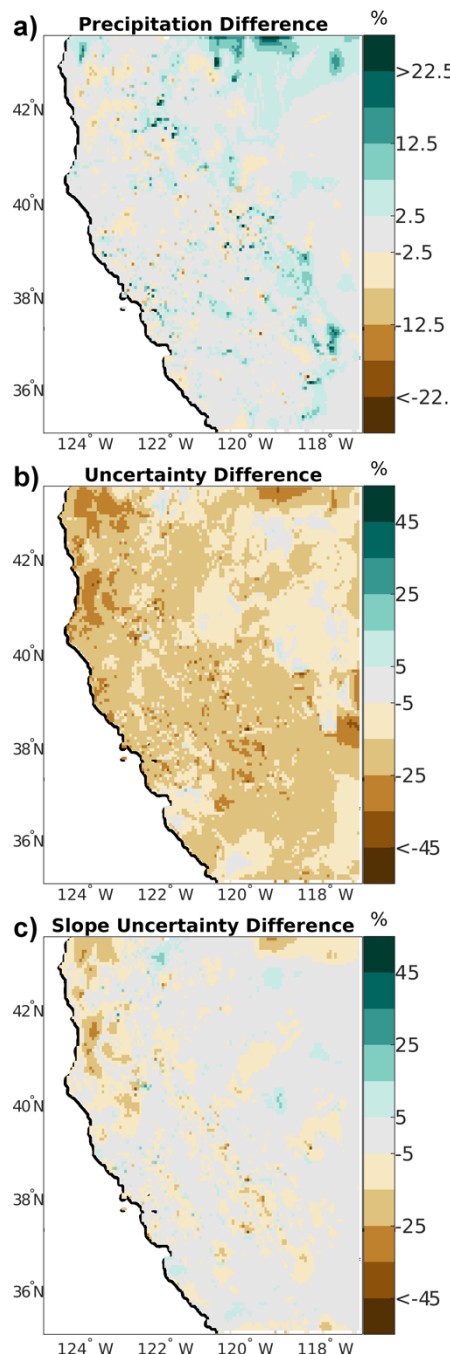

**Figure 11: Spatial distribution of differences for model parameter sensitivity experiment 3, modified maximum number of stations parameter (nMaxNear). a) precipitation differences (%), b) total uncertainty differences (%), and c) uncertainty changes due to the slope term in the regression.**

**Tables**

Table 1. Terrain preprocessing model parameters.

| Parameter | Default Value | Brief Description |
|---|---|---|
| demFilterName | Daly | Terrain Filter type (Daly = original Daly et al. 1994 filter)[1] |
| demFilterPasses | 8 | Number of passes to filter raw DEM |
| minGradient | 0.003 m km$^{-1}$ | Minimum gradient for a grid point to be considered sloped; otherwise it is considered flat |
| smallFacet | 500 km$^2$ | Area of smallest sloped facet allowed |
| smallFlat | 1000 km$^2$ | Area of smallest flat facet allowed |
| narrowFlatRatio | 3.1 | Ratio of major/minor axes to merge flat regions (e.g. ridges) |
| layerSearchLength | 10 grid points | Search length to determine local minima in elevation |
| inversionHeight | 250 m | Depth of atmospheric layer 1 (inversion layer) |

5   [1]Only filter option currently implemented

Table 2. Terrain preprocessing control file.

| Variable | Value | Brief Description |
|---|---|---|
| rawGridName | /path/to/input/raw/grid/file | Raw domain DEM |
| outputGridName | /path/to/output/processed/grid/file | Name of output processed grid |
| stationPrecipPath | /path/to/precipitation/station/data/directory | Path to precipitation station data |
| stationPrecipListName | /path/to/precipitation/metadata/output/file | Name of generated precipitation station list file |
| stationTempPath | /path/to/temperature/station/data/directory | Path to temperature station data |
| stationTempListName | /path/to/temperature/metadata/output/file | Name of generated temperature station list file |
| preprocessParameterFile | /path/to/TIER/preprocessing/parameter/file | Name of TIER preprocessing parameter file |

Table 3. TIER model parameters. Default values are given for precipitation with values for temperature given in parentheses.

| Parameter | Default Value | Brief Description |
|---|---|---|
| nMaxNear | 10 | Maximum number of nearby stations to consider |
| nMinNear | 3 | Minimum number of nearby stations needed for slope regression |
| maxDist | 250 km | Maximum distance to consider stations |
| minSlope | 0.25 (-10 K km$^{-1}$) | Minimum valid slope value (normalized for precipitation; physical units for temperature) |
| maxInitialSlope | 4.25 | Maximum valid initial pass normalized slope for precipitation |
| maxFinalSlope | 3.0 | Maximum valid final adjusted normalized slope for precipitation |
| maxSlopeLower | 20 K km$^{-1}$ | Maximum valid slope for temperature in lower atmospheric layer (inversion layer; allows for strong inversions) |
| maxSlopeUpper | 0 K km$^{-1}$ | Maximum valid slope for temperature in upper layer (free atmosphere; up to isothermal allowed) |
| defaultSlope | 1.3 (-6.5 K km$^{-1}$) | Default slope value (normalized for precipitation; physical units for temperature) |
| topoPosMinDiff | 500 m | Minimum elevation difference used to adjust topographic position weights |
| topoPosMaxDiff | 5000 m | Maximum elevation difference for stations to receive topographic position weighting |
| topoPosExp | 1.0 | Exponent in topographic position weighting function |
| coastalExp | 0.75 | Exponent in distance to coast weighting function |
| layerExp | 0.5 | Exponent in atmospheric layer weighting function |
| distanceWeightScale | 16000 | Scale parameter in Barnes (1964) distance weighting function |
| distanceWeightExp | 2 | Exponent in Barnes (1964) distance weighting function |
| maxGrad | 2.5 | Maximum allowable normalized precipitation slope gradient between grid cells |
| bufferSlope | 0.02 | Buffer parameter when computing precipitaiton slope feathering |
| minElev | 100 m | Minimum elevation considered when feathering precipitation |
| minElevDiff | 500 m | Minimum elevation difference across precipitation considered for feathering precipitation |
| recomputeDefaultPrecipSlope | True | Logical string to indicate re-estimation of the default slope using domain specific information |
| recomputeDefaultTempSlope | True | Logical string to indicate re-estimation of the default slope using domain specific information |
| filterSize | 15 grid points | Size of low pass filter used in computing updated slopes and uncertainty estimates |
| filterSpread | 11 | Spread of low-pass filter power used in computing updated slopes and uncertainty estimates |
| covWindow | 10 grid points | Window for local covariance calculation for the SYMAP and slope uncertainty components. Used in the final uncertainty estimation routine |

Table 4. TIER model control file.

| Variable | Value | Brief Description |
|---|---|---|
| gridName | /path/to/grid/file | Domain file name |
| variableEstimated | precip (tmax, tmin) | Name of meteorological variable estimate |
| stationFileList | /path/to/station/list/file | Name of variable specific (e.g. precip or tmax/tmin) file with list of input station files |
| stationDataPath | /path/to/station/data/directory | Path to station data |
| outputName | /path/to/output/file | Name of output file |
| parameterFile | /path/to/TIER/parameter/file | Name of TIER parameter file |
| defaultTempLapse | /path/to/default/temperature/lapse/rate/file | Name of default temperature lapse rate file |

Table 5. Calibration sample evaluation statistics for TIER using the default parameters with 90% confidence intervals in parenthesis.

|  | Precipitation (mm) | Maximum Temperature (K) | Minimum Temperature (K) |
|---|---|---|---|
| Bias | 0.2 (-1.1 – 1.5) | -0.22 (-0.28 – -0.15) | -0.21 (-0.28 – -0.16) |
| MAE | 14.3 (13.3 – 15.3) | 0.84 (0.79 – 0.90) | 0.75 (0.71 – 0.79) |

Table 6. Calibration sample evaluation statistics for the three experiments.

| | Experiment 1 Minimum Temperature (K) | Experiment 2 Precipitation (mm) | Experiment 3 Precipitation (mm) |
|---|---|---|---|
| Bias | -0.19 (-0.26 – -0.13) | 0.2 (-1.0 – 1.3) | 0.2 (-1.2 – 1.7) |
| MAE | 0.81 (0.77 – 0.86) | 12.5 (11.6 – 13.5) | 15.1 (14.1 – 16.1) |