# Peer review of "TIER Version 1.0: An open-source Topographically InformEd Regression (TIER) model to estimate spatial meteorological fields"

_Geoscientific Model Development, 2019_

## Referee Comment (RC1) · Anonymous Referee #1 · 19 Sep 2019

As described in the paper, the TIER model seems to emulate much of what Chris Daly's group has implemented in PRISM. The value of the paper and the accompanying code is thus 2-fold: first, it concisely summarizes the various parameterizations and assumptions in PRISM that have been published in a series of papers over a number of years, and second, it is open-source, allowing anyone to change or replace the parameterizations and assumptions as they see fit. Thus, while it's hard to identify anything in TIER that is original, I think the model and the accompanying paper are both valuable contributions worthy of prompt publication.

---

## Referee Comment (RC2) · Anonymous Referee #2 · 30 Sep 2019

This is an excellent technical manuscript on a rather comprehensive rainfall and temperature interpolation procedure that can benefit many scientific users, especially with the freely available code.

I recommend publication, subject to the following improvements:

Major: 1. Pg 1 line 24: The review of methods are limited and should include more recent literature (last 5-10 years). Furthermore, the authors could be more critical of their proposed method, in consideration of many other evolving interpolation approaches.

2. Page 8, line 19-24: "In TIERv1.0 we have chosen to use the base grid point estimate, ub as the intercept value in the variable-elevation regression ....Therefore, we fully disassociate the intercept and slope estimates. This methodological choice should be examined in future work". I believe it is justified that this methodological choice be examined as part of the current manuscript, considering that it is the first time it is introduced.

3. The scientific contribution of this paper can be improved with a more integrative look at the parameter uncertainty across the different experiments. The authors should consider combining Figures 8-15 into 2-3 more summative figures and highlighting the relative uncertainty contributed by the different parameter assumptions. Furthermore, although there are brief mentions of complex terrain and dry areas in the discussion of the results, these are few. Spatial features of the interpolation results and uncertainty can be better discussed.

Minor:

Section 2.1 and 2.2.3 are unclear in the definition of the topographic position concept, what it signifies, how it affects inversion, and how it is being calculated. The authors refer to D94 D02 and D08, but I suggest an explicit introduction be included for completeness. A very brief explanation is given later in 2.3.1.4; this should be brought earlier in the text.

Page 6, line 12: "downweigh" instead of "down weight"

Page 7, line 30: "could impact the final interpolation in unexpected ways" is vague. Please include specifics.

Page 7, line 9: "weigh" instead of "weight"

Page 12, line 20: "Finally, note the total uncertainty is nearly unchanged (not shown)". Show or remove statement.

Page 13, line 15: Remove "Interestingly".

[Figure]

Page 14, line 15: "including true out of sample station networks" - please clarify.

---

## Author Comment (AC1) · 21 Nov 2019

Within our author comments, the original reviewer statements will be in black while our replies will be in red.

**Author Reply to Reviewer #1.**

As described in the paper, the TIER model seems to emulate much of what Chris Daly's group has implemented in PRISM. The value of the paper and the accompanying code is thus 2-fold: first, it concisely summarizes the various parameterizations and assumptions in PRISM that have been published in a series of papers over a number of years, and second, it is open-source, allowing anyone to change or replace the parameterizations and assumptions as they see fit. Thus, while it's hard to identify anything in TIER that is original, I think the model and the accompanying paper are both valuable contributions worthy of prompt publication.

We thank the reviewer for their very positive review of this paper. As the reviewer and our manuscript notes, this paper does not introduce new methodologies but does synthesize a lengthy list of literature from the PRISM algorithm and provides an open-source code base for experimentation. We believe this provides value to the community through the two reasons the reviewer states.

**Author Reply to Reviewer #2.**

This is an excellent technical manuscript on a rather comprehensive rainfall and temperature interpolation procedure that can benefit many scientific users, especially with the freely available code. I recommend publication, subject to the following improvements:

We thank this reviewer for their positive review and helpful comments. We hope this paper is a useful summary of one approach to knowledge-based meteorological interpolation following the PRISM algorithm, and the code base becomes useful for experimenting with the many methodological choices and parameters within this interpolation system. We believe the revised manuscript will satisfy the reviewer and be useful to the community.

Major: 1. Pg 1 line 24: The review of methods are limited and should include more recent literature (last 5-10 years). Furthermore, the authors could be more critical of their proposed method, in consideration of many other evolving interpolation approaches.

We will add additional discussion of other interpolation methods and how the knowledge based approach fits within the full complement of methods. This will provide improved linkages to the interpolation literature and be helpful to novice users. However, the main point of this paper is not to dissect which general method is better at interpolating meteorological variables, but to present a synthesis of a knowledge-based interpolation approach, so we plan to limit discussion of 'pros' and 'cons' of the various methods.

2. Page 8, line 19-24: "In TIERv1.0 we have chosen to use the base grid point estimate, ub as the intercept value in the variable-elevation regression ....Therefore, we fully disassociate the

intercept and slope estimates. This methodological choice should be examined in future work". I believe it is justified that this methodological choice be examined as part of the current manuscript, considering that it is the first time it is introduced.

We agree with this comment and will provide some additional analysis through one (1) additional figure to examine how changes in determining 'ub' change grid point estimates of precipitation and temperature.

3. The scientific contribution of this paper can be improved with a more integrative look at the parameter uncertainty across the different experiments. The authors should consider combining Figures 8-15 into 2-3 more summative figures and highlighting the relative uncertainty contributed by the different parameter assumptions. Furthermore, although there are brief mentions of complex terrain and dry areas in the discussion of the results, these are few. Spatial features of the interpolation results and uncertainty can be better discussed.

We agree with this comment and will reexamine Figures 8-15 to reduce the number of figures and enhance the impact of fewer, more integrated figures and discussion of the parameter experiments.  We will also add further discussion of changes in the spatial features of the interpolation results in the context of the complex terrain in the example domain.

Minor:
Section 2.1 and 2.2.3 are unclear in the definition of the topographic position concept, what it signifies, how it affects inversion, and how it is being calculated. The authors refer to D94 D02 and D08, but I suggest an explicit introduction be included for completeness. A very brief explanation is given later in 2.3.1.4; this should be brought earlier in the text.

We will modify sections 2.1 and 2.2.3 by moving the text from 2.3.1.4 to the earlier sections and also add additional explanation of the topographic position calculation and what it is useful for.

Page 6, line 12: "downweigh" instead of "down weight"
Page 7, line 30: "could impact the final interpolation in unexpected ways" is vague.
Please include specifics.
Page 7, line 9: "weigh" instead of "weight"
Page 12, line 20: "Finally, note the total uncertainty is nearly unchanged (not shown)".
Show or remove statement.
Page 13, line 15: Remove "Interestingly".
Page 14, line 15: "including true out of sample station networks" - please clarify.

We have reviewed this list of minor comments and will make the suggested changes and clarifications in the revised manuscript.